# Overview of Mathematical Relations Between Poincaré Plot Measures and Time and Frequency Domain Measures of Heart Rate Variability

**DOI:** 10.3390/e27080861

**Published:** 2025-08-14

**Authors:** Arie M. van Roon, Mark M. Span, Joop D. Lefrandt, Harriëtte Riese

**Affiliations:** 1Department of Vascular Medicine, University of Groningen, University Medical Center Groningen, 9700 RB Groningen, The Netherlands; j.d.lefrandt@umcg.nl; 2Department of Experimental Psychology, University of Groningen, 9712 TS Groningen, The Netherlands; m.m.span@rug.nl; 3Department of Psychiatry, Interdisciplinary Center Psychopathology and Emotion regulation (ICPE), University of Groningen, University Medical Center Groningen, 9700 RB Groningen, The Netherlands; h.riese@umcg.nl

**Keywords:** heart rate variability, mathematical comparison, Poincaré plot

## Abstract

The Poincaré plot was introduced as a tool to analyze heart rate variations caused by arrhythmias. Later, it was applied to time series with normal beats. The plot shows the relationship between the inter-beat interval (IBI) of one beat to the next. Several parameters were developed to characterize this relationship. The short and long axis of the fitting ellipse, *SD*1 and *SD*2, respectively, their ratio, and their product are used. The difference between the IBI of a beat and m beats later are also studied, *SD*1(*m*) and *SD*2(*m*). We studied the mathematical relations between heart rate variability measures and the Poincaré measures in the time (standard deviation of IBI, *SDNN*, root mean square of successive differences, *RMSSD*) and frequency domain (power in low and high frequency band, and their ratio). We concluded that *SD*1 and *SD*2 do not provide new information compared to *SDNN* and *RMSSD*. Only the correlation coefficient *r*(*m*) provides new information for *m* > 1. Novel findings are that *ln*(*SD*2(*m*)/*SD*1(*m*)) = *tanh*^−1^(*r*(*m*)), which is an approximately normal distributed transformation of *r*(*m*), and that *SD*1(*m*) and *SD*2(*m*) can be calculated by multiplying the power spectrum by a weighing function that depends on *m*, revealing the relationship with spectral measures, but also the relationship between *SD*1(*m*) and *SD*2(*m*). Both lagged parameters are extremely difficult to interpret compared to low and high frequency power, which are more closely related to the functioning of the autonomic nervous system.

## 1. Introduction

Indices of the cardiovascular system have been studied for many purposes and in various ways, such as to predict patients’ severity or incidence of cardiovascular disorders [1], as well to investigate mechanisms associated with cognitive or psychological processes [2,3]. In particular, heart rate dynamics have led to the development of numerous methods for calculating heart rate variability (HRV). Heart rate variability refers to the variation in time intervals between heartbeats (inter-beat interval, IBI), which are regulated by the autonomic nervous system (ANS). HRV measures reflect the functioning of the ANS [1,2,3].

A landmark paper by a cardiology Task Force [1] provided a comprehensive overview of the significance and calculation of HRV measures. They categorized these measures into two groups, time and frequency domain measures.

Time domain measures that are calculated from the IBI time series are the standard deviation of the normal-to-normal interval (*SDNN*) and the root mean square of successive differences (*RMSSD*). Measures that are derived from the power density spectrum of IBI are the power in the low frequency band (*LF*, 0.04–0.14 Hz), power in the high frequency band (*HF*, 0.15–0.40 Hz), and their ratio *LF/HF*.

An update of the Task Force overview was made by Shaffer and Ginsberg [2], highlighting renewed interest in another approach towards HRV characterization, the Poincaré plot. This graph visualizes the relationship between the inter-beat interval of beat *i* (*IBI_i_*, on the *x*-axis) and the inter-beat interval of the next beat *i* + 1 (*IBI_i+_*_1_, on the *y*-axis). The Poincaré plot is used to visualize and analyze the behavior of a nonlinear dynamic system [4]. Its use is not limited to IBI time series; it can be used to analyze any time series [5,6,7,8,9].

The Poincaré plot was introduced by Woo [10] as a tool to analyze heart rate variations caused by arrhythmias. Time series with arrhythmias show a plot with several clusters (e.g., Figure 1A). Most types of arrhythmias have a distinct IBI pattern. It starts with a short interval—the ectopic beat occurs earlier than the expected normal beat, and this beat is followed by a longer interval—and the expected normal beat would start during the phase that the heart is still recovering from the ectopic beat.

For time series with normal beats, the shape of the plot is ellipse-like. The short and long axes of the characteristic ellipse are *SD*1 and *SD*2, respectively (Figure 1B). The long axis of the ellipse coincides with the line *y* = *x*, while the middle point of the ellipse is the point with an *x*- and *y*- coordinate equal to the mean inter-beat interval. In some publications [11,12], the plot is called a Lorenz plot with a *T* (*SD*1) and *L* (*SD*2) axis. Additionally, combinations of the parameters were studied. Following the *LF/HF* ratio in the frequency domain, which was originally intended [2] as a parameter to reflect the balance between parasympathetic and sympathetic parts of the ANS, ratios [13,14] of *SD*1 and *SD*2, its product [15,16], which is related to the area of the ellipse, and the ratio of the standard deviation of the successive differences of IBI (*SDSD*) and *SDNN* are introduced.

During the 1990s, the physiological interpretation of *SD*1 and *SD*2 was developed with the idea that these two parameters could show different aspects of HRV than *SDNN* and *RMSSD*. However, Brennan [17] showed that this is not the case. However, this notion was not widely adopted by authors; they were still treated as independent parameters by many. For instance, several studies [12,15,18] showed a very high correlation (0.98–1.00) between *SD*1 and *RMSSD*, although sometimes the correlation dropped [12] to 0.65 during exercise with a complete parasympathetic blockade. A book on this topic [19] was published in 2013, and a reminder was published in 2017 by Ciccone [20] about the equivalence of *RMSSD* and *SD*1 and that it is not worthwhile to report them both.

Beside the differences between two successive inter-beat intervals, the difference between inter-beat interval of a beat a number m beats later, *IBI_i+m_*, and inter-beat interval is studied. The corresponding Poincaré plot is named the extended [5] or *m*-lagged [21] plot, with *IBI_i+m_* on the *y*-axis. While the parameters of lag 1 are related to fast, parasympathetic changes at larger lags, slower and longer lasting, sympathetic effects can be studied. Moreover, the relationship of the Poincaré parameters with spectral measures is studied [12,15,21], showing different correlations and interpretations. Contreras [21] studied the relationship between *LF*, *HF*, and *SD*1(*m*) for *m =* 1 …10. The correlation was different for the different lags and differed between healthy controls and diabetic patients.

This paper is divided into two parts. In part I—named ‘Time Domain’—we summarize Brennan’s publication [17], in which he uses the expected value function to show the relationship between the Poincaré plot parameters. We explore the mathematic relationships of *SD*1, *SDSD*, and *RMSSD* in full detail. The introduced ratios and area of the ellipse for different lags are evaluated, using the autocorrelation equations of Brennan as a starting point.

In part II—named ‘Frequency Domain’—we investigate how the Poincaré parameters are related to the distribution of power in the spectrum. We investigate the relationship between *m*-lagged parameters *SD*1(*m*) and *SD*2(*m*) and the power density spectrum in the second part. We approach the *m*-lag Poincaré plot as a filtering process of the IBI time series. This will reveal the close relationship with the power density spectrum of IBI and the *m*-lagged Poincaré measures.

In this manuscript, we focus on the mathematical relationships between the IBI measures and extend the calculations to *m*-lagged derived parameters, giving a comprehensive overview of the Poincaré plot analysis in the time and frequency domain. Moreover, the results can be used for other time series as well.

## 2. Materials and Methods

We summarized the symbols used and their definitions in the abbreviation section.

### 2.1. Tools for Part I: Time Domain

In this methods section, we give an overview of some basic mathematics that we need to derive the relationships described by Brennan [17] and new relationships for the derived measures, such as *SD*1/*SD*2. We use the expected value of a time series variable *x*(*t*), having *N* equidistant samples *x_i_* at *t_i_*, which is defined as follows [17,22]:(1)$$E\left[x(t)\right]=\frac{1}{N}\sum_{i=1}^{N}x_{i}=\bar{x}$$
with basic rule *E[x + y] = E[x] + E[y]*. The variance of *x*(*t*) is shown as follows [17,22]:(2)$$E\left[{\left(x\left(t\right)-\bar{x}\right)}^{2}\right]=E\left[x^{2}\left(t\right)\right]-\;{\bar{x}}^{2}$$ The autocorrelation *R*(*τ*) for a time series *x*(*t*) is shown as follows [17,22]:(3)$$R\left(\tau \right)=E\left[\left(x\left(t\right)-E[x\left(t\right)]\right)\left(x\left(t+\tau \right)-E[x\left(t+\tau \right)]\right)\right]$$
and for a stationary time series, $E\left[x\left(t\right)\right]=E\left[x\left(t+\tau \right)\right]=\bar{x}$ [22]. *R*(*τ*) is then found with the following equation [17]:(4)$$R\left(\tau \right)=E\left[x\left(t\right)x\left(t+\tau \right)\right]-{\bar{x}}^{2}$$

### 2.2. Tools for Part II: Frequency Domain

In the frequency domain analysis, we show that *SD*1(*m*) and *SD*2(*m*) can be calculated as the output of a filter applied to the IBI time series. For this filter approach, we use the following basic summation rule for integration:(5)$$\int f\left(x\right)+g\left(x\right)\;dx=\int f\left(x\right)\;dx+\int g\left(x\right)\;dx$$

The autocorrelation function is the inverse Fourier transform of the power density spectrum, also known as the Wiener–Khinchine relation [22,23]. For single-sided power density spectra of IBI, *P_IBI_*(*f*), the relation is shown as follows [22]:(6)$$R\left(\tau \right)=\int_{0}^{\infty }P_{IBI}\left(f\right)\cos\left(2\pi f\tau \right)df$$

With these tools, we establish the relationships between the indices mathematically.

## 3. Results

### 3.1. Part I: Time Domain

#### 3.1.1. The Expected Value Approach (Brennan’s Findings)

In this section, we summarize the findings of Brennan [17]. The time series of inter-beat intervals IBI is used to calculate the time series of successive differences (SD). Usually, the differences with the next beat are calculated, but larger beat lags are also used. We refer to this time series as successive differences with a lag of *m* beats, as follows:(7)$${SD}_{i}\left(m\right)={IBI}_{i+m}-{IBI}_{i}$$ We calculate the statistics for *N* inter-beat intervals, and thus *N* − 1 successive intervals, but assume that *N* + *m* inter-beat intervals are available. For large *N* and a stationary IBI time series, the mean of the successive differences will be very close to zero (this is explained in the next section in detail). The mean and standard deviation can be defined by the following expected value function:(8)$$\bar{IBI}=E\left[{IBI}_{i}\right]\;$$(9)$$\bar{SD(m)}=E\left[{SD}_{i}\left(m\right)\right]\approx 0$$(10)$${SDNN}^{2}=E\left[{IBI}_{i}^{2}\right]-{\bar{IBI}}^{2}$$(11)$${SDSD}^{2}\left(m\right)=E\left[{SD}_{i}^{2}\left(m\right)\right]-{\bar{SD(m)}}^{2}\approx E[{SD}_{i}^{2}\left(m\right)]$$
and by the following definition:(12)$${RMSSD}^{2}=E[{SD}_{i}^{2}\left(1\right)]$$ The autocorrelation or autocovariance function between the IBI time series and the *m*-lagged time series is shown as follows:(13)$$R\left(m\right)=E\left[\left({IBI}_{i+m}-\bar{IBI}\right)\left({IBI}_{i}-\bar{IBI}\right)\right]=E\left[{IBI}_{i+m}{IBI}_{i}\right]-{\bar{IBI}}^{2}$$ Clearly, when *m* = 0, Equation (13) is equal to (10), so the following is true:(14)$${SDNN}^{2}=R(0)$$ Now, for a stationary IBI time series, we can express *SDSD*^2^(*m*) as a difference of two autocorrelations, as follows:(15)$$\begin{array}{l} {SDSD}^{2}\left(m\right)=E\left[{\left({IBI}_{i+m}-{IBI}_{i}\right)}^{2}\right]\\ \;\;\;\;\;\;\;=E\left[{IBI}_{i+m}^{2}\right]-{\bar{IBI}}^{2}-2E\left[{IBI}_{i+m}{IBI}_{i}\right]+2{\bar{IBI}}^{2}+E\left[{IBI}_{i}^{2}\right]-{\bar{IBI}}^{2}\\ \;\;\;\;\;\;\;=2(R(0)-R(m)) \end{array}$$ The Pearson correlation coefficient for lag *m* is shown as follows [17,22]:(16)$$r\left(m\right)=\frac{E[\left({IBI}_{i+m}-\bar{IBI}\right)\left({IBI}_{i}-\bar{IBI}\right)]}{\sqrt{E[{\left({IBI}_{i+m}-\bar{IBI}\right)}^{2}]E[{\left({IBI}_{i}-\bar{IBI}\right)}^{2}]}}=\frac{R(m)}{R(0)}$$ Brennan [17] showed that the short and long axes of the ellipse in the lag *m* Poincaré plot can be found as follows:(17)$${SD1}^{2}\left(m\right)=R\left(0\right)-R\left(m\right)=\frac{1}{2}{SDSD}^{2}\left(m\right)$$(18)$${SD2}^{2}\left(m\right)=R\left(0\right)+R\left(m\right)=2{SDNN}^{2}-\frac{1}{2}{SDSD}^{2}(m)$$
*SD*1(*m*) is the standard deviation of the time series *SD*1*_i_*(*m*), as defined in Equation (7), divided by *√*2. Therefore, the sum of *SD*1^2^(*m*) and *SD*2^2^(*m*) can be found as follows:(19)$${SD1}^{2}\left(m\right)+{SD2}^{2}\left(m\right)=2R\left(0\right)=2{SDNN}^{2}$$ This means that, for all *m*, *SD*1 or *SD*2 is sufficient to report if *SDNN* is reported. The geometric representation of Equation (19) is that *SDNN √*2 is equal to the hypotenuse of a right triangle with sides *SD*1(*m*) and *SD*2(*m*). This is indicated in Figure 1B for *m* = 1. If *R*(*m*) becomes zero, and thus the correlation *r*(*m*) = 0, we find the following to be true:(20)$$r\left(m\right)=0\Rightarrow SD1\left(m\right)=SD2\left(m\right)=SDNN$$

#### 3.1.2. Relation *RMSSD*, *SDSD*, and *SD*1

As shown in the previous section, the differences between *RMSSD*, *SDSD*, and *SD*1 are only small. We show the precise relationships in this section for the ‘original’ successive differences (*m* = 1), in order to show under which conditions the trio can be considered to be identical. The *N* inter-beat intervals result in *N −* 1 successive differences with mean, and the equation is shown as follows:(21)$$\bar{SD}=\frac{1}{N-1}\sum_{i=1}^{N-1}{SD}_{i}=\frac{1}{N-1}\left[\sum_{i=1}^{N-1}{IBI}_{i+1}-\sum_{i=1}^{N-1}{IBI}_{i}\right]=\frac{1}{N-1}\left[\sum_{i=2}^{N}{IBI}_{i}-\sum_{i=1}^{N-1}{IBI}_{i}\right]$$ We make the summation range equal for both summations, but we have to subtract the extra values as follows:(22)$$\bar{SD}=\frac{1}{N-1}\left[\left(\sum_{i=1}^{N}{IBI}_{i}-{IBI}_{1}\;\right)-\left(\sum_{i=1}^{N}{IBI}_{i}-{IBI}_{N}\right)\right]=\frac{{IBI}_{N}-{IBI}_{1}}{N-1}$$ For large *N* and/or small difference between the first and last inter-beat interval, the mean of SD will be close to zero. If a time series is stationary, assumed for all time domain and spectral measures, the difference between the first and last beat will indeed be small. Now, the squared parameters are shown as follows:(23)$${RMSSD}^{2}=\frac{1}{N-1}\sum_{i=1}^{N-1}{SD}_{i}^{2}=\left\{\begin{array}{c} {SDSD}^{2}+{\bar{SD}}^{2}\\ 2{SD1}^{2}+{\bar{SD}}^{2} \end{array}\right.$$(24)$${SDSD}^{2}=\frac{1}{N-1}\sum_{i=1}^{N-1}{\left({SD}_{i}-\bar{SD}\right)}^{2}=\left\{\begin{array}{c} {RMSSD}^{2}-{\bar{SD}}^{2}\\ 2{SD1}^{2} \end{array}\right.$$(25)$${SD1}^{2}=\frac{1}{N-1}\sum_{i=1}^{N-1}{\left(\frac{{SD}_{i}-\bar{SD}}{\sqrt{2}}\right)}^{2}=\left\{\begin{array}{c} \frac{1}{2}\left({RMSSD}^{2}-{\bar{SD}}^{2}\right)\\ \frac{1}{2}{SDSD}^{2} \end{array}\right.$$ Note that we divided the sum by *N* − 1 instead of *N* − 2 for *SDSD* and *SD*1, which corresponds with the expected value approach of Brennan [17]. When $\bar{SD}\approx 0$, then this reduces to the following:(26)$${RMSSD}^{2}={SDSD}^{2}=2{SD1}^{2}$$ For stationary IBI time series of normal beats with sufficient length (usually 5 min or more), this is valid.

#### 3.1.3. Derived Measures *SD*1/*SD*2, *SDSD*/*SDNN*, and Ellipse Area

A number of measures are derived from the time domain measures and *SD*1 and *SD*2, including the ratio of *SD*1 and *SD*2 (and of *SD*2 and *SD*1), the ratio *SDSD/SDNN*, and the area of the ellipse. The rationale for the use of ratios is the same as for *LF/HF* in the frequency domain. However, now aware of the relationship between *SD*1, *SD*2, *SDSD*, and *SDNN*, the relationship between the derived measures can be explored, not only for lag *m* = 1, but for any lag. We start with the most commonly used ratios, *SD*1/*SD*2 and *SD*2/*SD*1, as follows:(27)$$\frac{{SD1}^{2}(m)}{{SD2}^{2}(m)}=\frac{R\left(0\right)-R(m)}{R\left(0\right)+R(m)}=\frac{1-R(m)/R(0)}{1+R(m)/R(0)}=\frac{1-r(m)}{1+r(m)}$$ So,(28)$$\frac{SD1(m)}{SD2(m)}=\sqrt{\frac{1-r(m)}{1+r\left(m\right)}}\;and\;\frac{SD2(m)}{SD1(m)}=\sqrt{\frac{1+r(m)}{1-r(m)}}$$ For *r*(*m*) values that are close to zero, (*|r*(*m*)*|* < 0.15), the ratio *SD*1/*SD*2 can be approximated by the following:(29)$$\frac{SD1(m)}{SD2(m)}=\sqrt{\frac{1-r(m)}{1+r\left(m\right)}\;\frac{1-r(m)}{1-r\left(m\right)}}=\left(1-r\left(m\right)\right)\sqrt{\frac{1}{1-r^{2}\left(m\right)}}\approx 1-r(m)$$ The ratios only depend on the correlation between the original and lagged IBI series. This relationship was already shown by Carrasco [12] for Lorenz plots (in Figure 2 of Carrasco, *L/T = SD*2(1)*/SD*1(1) is shown in relation to positive *r*(*m*) values). Interestingly, the natural logarithm of *SD*2/*SD*1 is commonly used as a normal distributed transformation of correlations coefficients, shown as follows [12,24,25]:(30)$$z(m)=ln\left(\frac{SD2\left(m\right)}{SD1\left(m\right)}\right)=ln\left(\sqrt{\frac{1+r\left(m\right)}{1-r\left(m\right)}}\right)=\frac{1}{2}ln\left(\frac{1+r\left(m\right)}{1-r\left(m\right)}\right)={tanh}^{-1}(r\left(m\right))$$
*Tanh*^−1^(*x*) is the inverse hyperbolic tangent function. Note that the range of this function is from −∞ (for *r*(*m*) = −1) to +∞ for (*r*(*m*) = 1) and that, if the value *z*(*m*) *= c* for *r*(*m*), then *z*(*m*) *= −c* for *−r*(*m*). Since *z*(*m*) and *SD*2(*m*)*/SD*1(*m*) are strictly monotonic increasing functions of *r*(*m*), the inverse functions exist, and the interpretations of *z*(*m*), *SD*2/*SD*1, and *r*(*m*) are identical. Taking the natural logarithm of *SD*1/*SD*2 but with a minus sign gives the same result.

For the ratio *SDSD/SDNN*, we find the following:(31)$$\frac{{SDSD}^{2}(m)}{{SDNN}^{2}}=\frac{2(R\left(0\right)-R\left(m\right))}{R(0)}=2\left(1-\frac{R\left(m\right)}{R\left(0\right)}\right)=2\left(1-r\left(m\right)\right)$$ Hence, the following can be found:(32)$$\frac{SDSD(m)}{SDNN}=\sqrt{2\left(1-r\left(m\right)\right)}$$ Rearranging (32) gives an easy way to calculate the correlation *r*(*m*), as follows:(33)$$r\left(m\right)=1-\frac{1}{2}\frac{{SDSD}^{2}(m)}{{SDNN}^{2}}$$

In Figure 2, the relationship between the ratios and *r*(*m*) is shown. Note that *r*(*m*) is a correlation coefficient and limited to the range −1 to 1. Because of this, the ratio *SDSD*(*m*)*/SDNN* is limited to the range 0 to 2 (and also the ratio *RMSSD/SDNN* is approximately limited to this range).

The last derived measure to explore is the area of the ellipse in the Poincaré plot. The area of an ellipse is the product of both axes multiplied by π. The squared product of *SD*1(*m*) and *SD*2(*m*) is shown as follows:(34)$$\begin{array}{l} {SD1}^{2}\left(m\right){SD2}^{2}\left(m\right)=\left(R\left(0\right)-R\left(m\right)\right)\left(R\left(0\right)+R\left(m\right)\right)=R^{2}\left(0\right)-R^{2}\left(m\right)\\ \;\;\;\;\;\;=R^{2}(0)\left(1-\frac{R^{2}(m)}{R^{2}(0)}\right) \end{array}$$ The area is now the following:(35)$$\pi \;SD1\left(m\right)\;SD2\left(m\right)=\pi \;R\left(0\right)\sqrt{1-{\left(\frac{R(m)}{R(0)}\right)}^{2}}=\pi \;{SDNN}^{2}\sqrt{1-r^{2}\left(m\right)}$$ The area values can range from 0 to *π SDNN*^2^. For time series with *r*^2^(*m*) ≈ 0.9, the area is equal to *SDNN*^2^. The ratio of the area and *SDNN*^2^ and *r*(*m*) is also shown in Figure 2. We have shown that all derived parameters are related to one new parameter, which is *r*(*m*). Reporting the correlation at lag *m* is therefore sufficient.

### 3.2. Part II: Frequency Domain

So far, we have evaluated the time domain measures. To explore the frequency domain, we assume that *SD*1^2^(*m*) or *SD*2^2^(*m*) is the total power of the output of a filter with the IBI time series as input. Applying a filter means, in the spectral domain, that you multiply the spectral values with certain weighing factors. An example is shown in Appendix B for the calculation of the power in the low frequency band *LF*. The filter characteristics determine the values of the weighing factors. In this section, we show which weighing factors must be used for a filter that has *SD*1^2^(*m*) and *SD*2^2^(*m*) as total power of the filter output. The filter characteristic *H*(*f*), the transfer function, of such a filter to calculate *SD*1(*m*) or *SD*2(*m*) must be determined.

The autocorrelation function is found as follows, using the Wiener–Khinchine relation (6), for lag *m*:(36)$$R\left(m\right)=\int_{0}^{\infty }P_{IBI}\left(f\right)\cos\left(2\pi fm\bar{IBI}/1000\right)df$$ We have to include *m* multiplied by $\bar{IBI}/1000$ to create an average time lag *τ* in seconds, instead of the lag in number of beats, in the cosine argument. Note that Parseval’s theorem is a special case of the Wiener–Khinchine relation (6); for *m* = 0, the cosine argument is always 0 and *cos*(0) = 1. For the correlation, we find the following:(37)$$r\left(m\right)=\;\frac{R(m)}{R(0)}=\frac{\int_{0}^{\infty }P_{IBI}\left(f\right)\cos\left(2\pi fm\bar{IBI}/1000\right)df}{\int_{0}^{\infty }P_{IBI}\left(f\right)\;df}$$ This relationship can be helpful to explain findings where a specific spectral shape is assumed, like in the simulations performed by Satti [5]. In Appendix C, we show this in detail. In Appendix D, we show an alternative approach to calculate the transfer function, which is more direct but requires knowledge of the mathematics of complex numbers.

Now, we start with *SD*1(*m*), combining Equations (17), (36), and (5) as follows:(38)$${SD1}^{2}\left(m\right)=R\left(0\right)-R\left(m\right)=\int_{0}^{\infty }P_{IBI}\left(f\right)df-\int_{0}^{\infty }P_{IBI}\left(f\right)\cos\left(2\pi fm\bar{IBI}/1000\right)df\;=\int_{0}^{\infty }(1-\cos\left(2\pi fm\bar{IBI}/1000\right)){\;P}_{IBI}\left(f\right)df$$ For *SD1*(*m*) squared, the transfer function is shown as follows:(39)$${\left|H_{SD1(m)}\left(f\right)\right|}^{2}=1-cos(2\pi fm\bar{IBI}/1000)$$ For *SD*2(*m*), the only difference is the minus sign, shown as follows:(40)$${\left|H_{SD2(m)}\left(f\right)\right|}^{2}=1+cos(2\pi fm\bar{IBI}/1000)$$

In Figure 3, we show the squared transfer functions for *SD*1(*m*), at lags *m =* 1 …6 and 10, in case mean IBI is 1000 ms, in the left panel. In the right panel, an example of IBI power density spectrum is shown (dashed). Multiplying the IBI power spectrum by the squared transfer function for each frequency, as described in Equation (38), results in the spectrum of *SD_i_*(*m*) (solid). The area under this spectrum equals the total power, which is *SD*1^2^(*m*). In Figure 4, the same panels for *SD*2(*m*) are shown. In the Appendix A, an Excel sheet is available to create these figures for other $\bar{IBI}$ and *m* values.

Note that, for *RMSSD*, the transfer function of *SD*1(1) can be used, multiplied by 2. In the example, at lag *m =* 5, the power of *LF + HF* will be close to *SD*1^2^(5). Increasing the lag to very high values will result in resampling the power density spectrum at a high frequency (*m*/2 samples). *SD*1(*m*) and *SD*2(*m*) will converge to *SDNN*; although, they will still include complementary frequencies. The ratio *SD*1*/*(*m*)*/SD*2(*m*) will converge to 1, corresponding with *r*(*m*) converging to 0, which makes sense, given the large lag between the beats and a non-stationary recording becoming more likely.

## 4. Discussion

We evaluated the mathematical relationships between time domain HRV measures *SDNN* and *RMSSD* and a series of Poincaré parameters, *SD*1, *SD*2, *SDSD*, their ratios, and their product, for different time lags. Like others [17,19,20], we show the great overlap between the time domain and Poincaré measures. We show that the correlation *r*(*m*) is the only parameter that adds new information to the time domain parameters for *m* > 1, which is also introduced in this way by Satti [5]. It is not surprising, since the Poincaré plot is in fact an analysis of the autocorrelation between the IBI time series and its m beats lagged time series. Usually [2], *SD*1 and *SD*2 are referred to as non-linear parameters, but the parameters are clearly of the same type as *SDNN* and *RMSSD* [4,17].

For larger values of *m*, the correlation between beat *i* and *i + m* becomes smaller. In that case, *SD*1(*m*) *≈ SD*2(*m*) *≈ SDNN*. Empirical data confirming this can be found in Figure 1 of Koichubekov [26] (left panel). At higher values of *m*, the *SD*1(*m*)*/SD*2(*m*) ratio is about 0.9 (right panel), which means that the correlation *r*(*m*) ≈ 0.1, using the approximation of *SD*1(*m*)*/SD*2(*m*) *≈* 1 − *r*(*m*). The ratio is often interpreted in the same way as *LF/HF*, and Guzik [15] shows a significant correlation between them. However, the interpretation of Stein [27], that it “captures the randomness of R-R-intervals”, is closer to what the ratio is, as *−ln*(*SD*1(*m*)*/SD*2(*m*)) is a normal distributed transformation of *r*(*m*). However, how can we explain the high correlation between *SD*2*/SD*1 (*m* = 1) and *LF/HF* that Guzik [15] found? Note that *SD*2/*SD*1 and *r*(1) are highly correlated (see Figure 2, over the range *r*(1) = 0.5 …0.8, *R*^2^ = 0.99) and that, for *r*(1), the spectral computation is found as follows:(41)$$r\left(1\right)=\frac{1}{{SDNN}^{2}}\int_{0}^{\infty }P_{IBI}\left(f\right)\cos\left(2\pi f\;\bar{IBI}/1000\right)df$$

For frequencies up to 0.25 Hz at mean IBI = 1000 ms (=1/4 1000$/\bar{IBI}$) the contribution of the power to *r*(1) is positive, but for higher frequencies, it is negative. Therefore, the major part of the *HF* power will reduce *r*(1), while *r*(1) will increase with increasing *LF* power. The power of *LF* and *HF* affects the ratio *LF/HF* in the same way, explaining their correlation.

Because of the close relationship between the Power Density Spectrum and the autocorrelation function *R*(*m*), by inverse Fourier transform, we could show the relationship between the m lagged parameters and the spectrum. Using a filter approach, we could determine the relationship between a given spectrum and the time domain and Poincaré parameters. The filter characteristic (squared modulus function) is determined by the lag m and the mean inter-beat interval of the time series. From this approach, we also know that *SD*1^2^ is closely related to *HF*, which affects the correlation between the ratios *SD*2*/SD*1 and *LF/HF*. *SD*2^2^ is mostly related to *VLF* and *LF* and can be used as an estimate for low frequency variability. However, while the frequency bands *VLF*, *LF*, and *HF* are independent, *SD*1^2^ and *SD*2^2^ are dependent parameters because of the overlap of their transfer functions shown in Figure 3 and Figure 4, in the range 0.15–0.30 Hz. Due to use of a lag expressed in beats, this range varies with mean IBI.

We limited our analysis to SD1, SD2, and a number of derived parameters and did not try to give a review of all publications on the subject, as they are so numerous. Of course, many other approaches used to quantify the visual and temporal aspects of Poincaré plots have been developed [19,28,29,30]. We think our findings can be helpful to determine whether (newly) developed measures provide new and independent information.

## 5. Conclusions

Applying the Poincaré plot for the detection and classification of ectopic beats, its intended use [10], is not questioned. However, for the analysis of normal heart rate variability, Poincaré plot measures at lag *m* = 1, *SD*1, and *SD*2 do not provide new information compared to *SDNN* and *RMSSD*.

We derived that *ln*(*SD*2(*m*)/*SD*1(*m*)) = −*ln*(*SD*1(*m*)/*SD*2(*m*)) is a normal distributed variable that can be used for statistical tests of the correlation coefficient *r*(*m*) and calculate confidence intervals for *r*(*m*). *SD*1(*m*) and *SD*2(*m*) are, in fact, frequency domain power measures with varying frequency bands. They can be calculated as the total power of the subjects power density spectrum after multiplying the appropriate transfer function for the desired lag. *SD*1^2^(1) as well as *RMSSD*^2^ are closely related to *HF*.

It is important to realize that the lagged parameters depend on mean IBI and the exact spectral density of a subject (most important here is the actual respiratory frequency), resulting in a weighing of the power density spectrum differently for each subject. *SD*2(*m*) is complementary to *SD*1(*m*) at any lag because their squared values always add up to 2 *SDNN*^2^. Both lagged parameters are extremely difficult to interpret compared to *LF* and *HF* powers, which are closer related to the functioning of the ANS.

All investigated Poincaré variables, *SD*1(*m*), *SD*2(*m*), *SD*2(*m*)*/SD*1(*m*), *SD*1(*m*)*/SD*2(*m*), *π × SD*1(*m*) *× SD*(2(*m*), and *SDSD*(*m*)*/SDNN*, are related to *r*(*m*). Reporting *r*(*m*) is sufficient to provide new information besides *SDNN*.

## Figures and Tables

**Figure 1 entropy-27-00861-f001:**
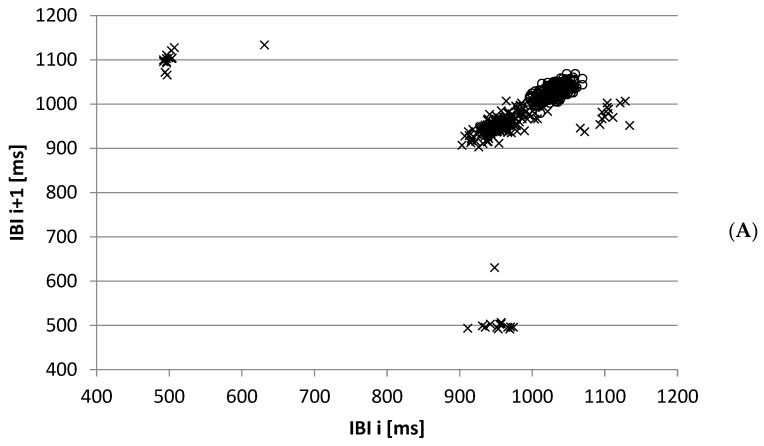
(**A**) Poincaré plot of a normal IBI time series (o) and with arrhythmias (x). (**B**) Normal IBI and ellipse with axes *SD*1 and *SD*2 and the hypotenuse (dashed) *SDNN √*2.

**Figure 2 entropy-27-00861-f002:**
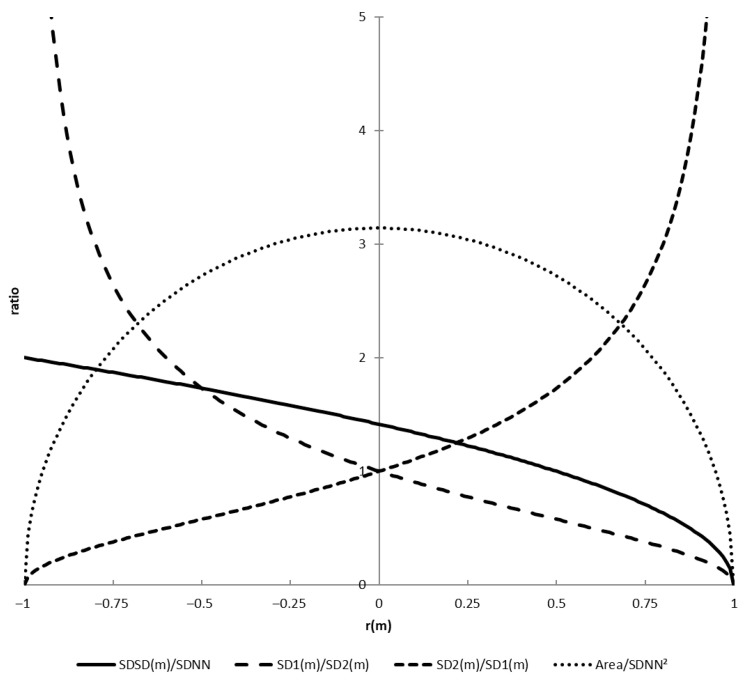
Relationship between correlation *r*(*m*) and the four ratios *SDSD*(*m*)*/SDNN*, *SD*1(*m*)*/SD*2(*m*), *SD*2(*m*)*/SD*1(*m*), and *Area/SDNN*^2^.

**Figure 3 entropy-27-00861-f003:**
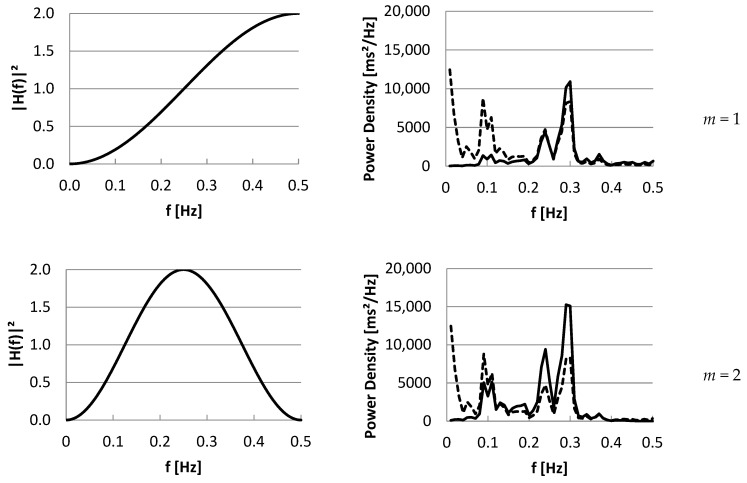
*SD*1(*m*) transfer functions (**left**) and an example of a spectral density function (**right**, dashed) for lags *m* = 1 …6 and 10 at mean IBI = 1000 ms. Multiplying the IBI power spectrum by the squared transfer function for each frequency, as described in Equation (38), results in the spectrum of *SD_i_*(*m*) (solid). The area under this spectrum equals the total power, which is *SD*1^2^(*m*). - - - P_IBI_(f) ____ P_SD1(m)_(f). In the Appendix A, an Excel sheet is available to create these figures for other $\bar{IBI}$ and *m* values.

**Figure 4 entropy-27-00861-f004:**
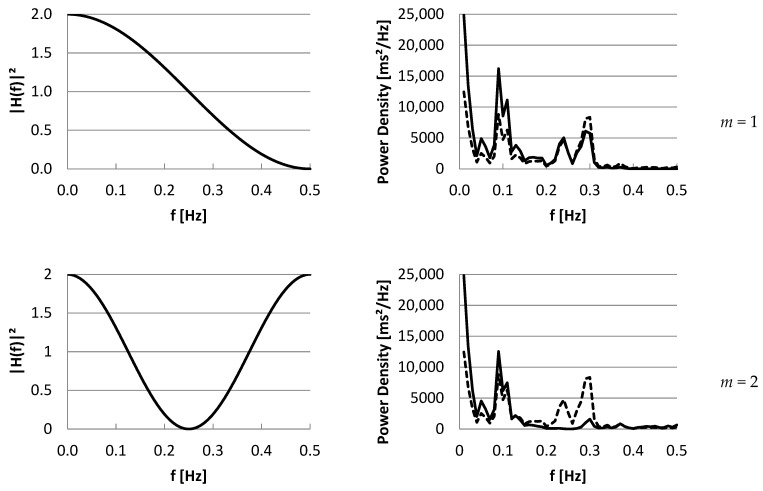
*SD*2(*m*) transfer functions (**left**) and an example of a spectral density function (**right**, dashed) for lags m = 1 …6 and 10 at mean IBI = 1000 ms. Multiplying the IBI power spectrum by the squared transfer function for each frequency, as described in Equation (40), results in the spectrum of the time series for *SD*2(*m*) (solid). The area under this spectrum equals the total power, which is *SD*2^2^(*m*). - - - P_IBI_(f) ____ P_SD2(m)_(f). In the Appendix A, an Excel sheet is available to create these figures for other $\bar{IBI}$ and *m* values.

## Data Availability

Data are contained within the article.

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
