# Peer review of "Overview of Mathematical Relations Between Poincaré Plot Measures and Time and Frequency Domain Measures of Heart Rate Variability"

_entropy, 2025, doi:10.3390/e27080861_

Round 1
Reviewer 1 Report
Comments and Suggestions for Authors
The authors explored the mathematical relationships between Poincaré plot measures (at different lags m) and time- and frequency-domain measures of heart rate variability (HRV). They showed that for lag m = 1, SD1 and SD2 are related to SDNN and RMSSD. SD1(m) and SD2(m) are also related to frequency-domain power measures with varying frequency bands. These findings align with the current understanding of the relationship between the classical Poincaré plot (m = 1). I believe the novel aspect of this study is twofold:
-
The relationship between extended Poincaré plot measures (with m>1) and time- and frequency-domain analyses is investigated.
-
This study correctly demonstrates that Poincaré plot measures are linear measures of HRV which is an important clarification, as these have often been mistakenly referred to as nonlinear measures in the literature. This misinterpretation typically arises from associating nonlinearity with the Poincaré plot method itself, rather than its specific measures (i.e., SD1, SD2, and r).
Based on this evaluation, I believe this study will be a valuable addition to the literature in this area.
I have the following comments that may help improve the study:
A. The authors introduce the Poincaré plot as if it is used exclusively for HRV assessment. However, the Poincaré plot and its associated measures were originally borrowed from the autocorrelation literature and have been widely used in physiology to analyse various types of time series, such as temperature variability (doi: 10.14814/phy2.14452), respiratory dynamics (reference 16), and oxygen saturation variability (doi: 10.3389/fphys.2017.00555). In these contexts, the Poincaré plot offers a simple method for quantifying short-term and long-term variability without requiring prior knowledge of the oscillatory nature or frequency characteristics of the time series. Therefore, its application is not limited to HRV. Moreover, the calculation of SD1 and SD2 is computationally simpler than frequency-domain analysis, which is advantageouses, specially in cases where distinct frequency bands are not well-defined for a given physiological signal. In such cases, while Poincaré plot measures may not provide additional information compared with frequency domain measures , they can still serve as simple indicators of short-term and long-term variability in complex time series.
B. The Methods section in its current form lacks clarity and focus. It primarily presents equations without adequately describing the methodological approach. This section could be improved by providing a summary of the overall methodology and detailing the data sources used in the study.
Author Response
The authors explored the mathematical relationships between Poincaré plot measures (at different lags m) and time- and frequency-domain measures of heart rate variability (HRV). They showed that for lag m = 1, SD1 and SD2 are related to SDNN and RMSSD. SD1(m) and SD2(m) are also related to frequency-domain power measures with varying frequency bands. These findings align with the current understanding of the relationship between the classical Poincaré plot (m = 1). I believe the novel aspect of this study is twofold:
The relationship between extended Poincaré plot measures (with m>1) and time- and frequency-domain analyses is investigated.
This study correctly demonstrates that Poincaré plot measures are linear measures of HRV which is an important clarification, as these have often been mistakenly referred to as nonlinear measures in the literature. This misinterpretation typically arises from associating nonlinearity with the Poincaré plot method itself, rather than its specific measures (i.e., SD1, SD2, and r).
Based on this evaluation, I believe this study will be a valuable addition to the literature in this area.
RESPONSE: Thank you for your review. Our adjustments in the text to your comments are highlighted in green.
I have the following comments that may help improve the study:
- The authors introduce the Poincaré plot as if it is used exclusively for HRV assessment. However, the Poincaré plot and its associated measures were originally borrowed from the autocorrelation literature and have been widely used in physiology to analyse various types of time series, such as temperature variability (doi: 10.14814/phy2.14452), respiratory dynamics (reference 16), and oxygen saturation variability (doi: 10.3389/fphys.2017.00555). In these contexts, the Poincaré plot offers a simple method for quantifying short-term and long-term variability without requiring prior knowledge of the oscillatory nature or frequency characteristics of the time series. Therefore, its application is not limited to HRV. Moreover, the calculation of SD1 and SD2 is computationally simpler than frequency-domain analysis, which is advantageouses, specially in cases where distinct frequency bands are not well-defined for a given physiological signal. In such cases, while Poincaré plot measures may not provide additional information compared with frequency domain measures, they can still serve as simple indicators of short-term and long-term variability in complex time series.
RESPONSE: We are well aware of the applications in other areas of physiology, and even outside that domain. It can be applied to any time series. However, HRV is our area of expertise and, to our knowledge, the domain where it is used most frequently. However, we want to acknowledge this important and relevant knowledge and added to the introduction: “Its use is not limited to IBI time series, it can be used to analyze any time series [5-9]” and “More-over, the results can be used for other time series as well”.
In the current manuscript, our major point is to show that SDNN, RMSSD, SD1 and SD2 are closely related and are not independent outcome measures. It is true that SD2 reflects low frequency variability and SD1 of high frequency variability and can be calculated without spectral analysis. However, this comes with a cost: (1) there is overlap in contribution of the mid frequency range around 0.25 Hz, as shown in the upper left panels (H(f), m=1) of figure 3 and 4; and (2) because of the use of lags expressed in beats, the frequency range is different if mean IBI is different. We added to the discussion: ” SD22 is mostly related to VLF and LF and can be used as an estimate for low frequency variability. However, while the frequency bands VLF, LF and HF are independent, SD12 and SD22 are dependent parameters because of the overlap of their transfer functions shown in figure 3 and 4, in the range 0.15-0.30 Hz. Due to use of a lag expressed in beats, this range varies with mean IBI.”
- The Methods section in its current form lacks clarity and focus. It primarily presents equations without adequately describing the methodological approach. This section could be improved by providing a summary of the overall methodology and detailing the data sources used in the study.
RESPONSE: We have added a summary and more references and divided the material and methods section in part I and II, as we did in the results section.
Reviewer 2 Report
Comments and Suggestions for Authors
The study mathematically explores the relationship between metrics derived from Poincaré plots and time or frequency domain Heart Rate Variability (HRV) metrics. As I lack expertise in evaluating mathematical equations, my review will primarily focus on the paper's presentation.
In Introduction:
The introduction is well-structured, and its presentation is both fluent and logical. Nevertheless, the significance of this research is not readily apparent.
In line 101, should "In his manuscript" be "In this manuscript"?
Applying italics to represent variables, parameters, or metrics in the text might be better.
In Materials and Methods
Variables in text should be italics. Such as N, x(t), xi, ti.
In equation 6, PIBI(f) did not be defined.
In Result:
Though the equation 17 have defined the SD12(m), could you kindly provide the definition of SD1 is the standard deviation of SDi(1).
In line 140, therefor or therefore?
It hard to understand the meaning of Fig.3 and 4. Could you kindly provide more explanation for readers who may lack related knowledge?
In conclusion
In line 283 Ln->ln.
Author Response
The study mathematically explores the relationship between metrics derived from Poincaré plots and time or frequency domain Heart Rate Variability (HRV) metrics. As I lack expertise in evaluating mathematical equations, my review will primarily focus on the paper's presentation.
RESPONSE: Thank you for taking the time for the review. The changes we made to your remarks are highlighted in yellow in the revised manuscript.
In Introduction
The introduction is well-structured, and its presentation is both fluent and logical. Nevertheless, the significance of this research is not readily apparent.
In line 101, should "In his manuscript" be "In this manuscript"?
RESPONSE: You are right, and we changed it accordingly.
Applying italics to represent variables, parameters, or metrics in the text might be better.
RESPONSE: We changed it throughout the manuscript, notably without highlighting all these changes.
In Materials and Methods
Variables in text should be italics. Such as N, x(t), xi, ti.
RESPONSE: We changed it throughout the manuscript, notably without highlighting all these changes.
In equation 6, PIBI(f) did not be defined.
RESPONSE: The definition is added.
In Results
Though the equation 17 have defined the SD12(m), could you kindly provide the definition of SD1 is the standard deviation of SDi(1).
RESPONSE: We added after eq 17: “SD1(m) is the standard deviation of the time series SDi(m) as defined in equation (7), divided by √2.”
In line 140, therefor or therefore?
RESPONSE: Changed into “therefore”.
It hard to understand the meaning of Fig.3 and 4. Could you kindly provide more explanation for readers who may lack related knowledge?
RESPONSE: We added more explanation in the text as well in the legend of figures 3 and 4. We hope this is sufficient: “Multiplying the IBI power spectrum by the squared transfer function for each frequency, as described in equation (38), results in the spectrum of SDi(m) (solid). The area under this spectrum equals the total power, which is SD12(m).” for SD1(m) and a similar text for SD2.
In conclusion
In line 283 Ln->ln.
RESPONSE: This is caused by the fact that it’s the beginning of a sentence. We changed it to “We derived that ln(…
Reviewer 3 Report
Comments and Suggestions for Authors
The study provides an overview of mathematical relations of Poincaré plot markers with time and frequency domain measures of heart period variability.
The study is interesting, even though its originality should be more clearly highlighted and discussion should be enlarged to account for the undoubtful usefulness of Poincaré plot indexes in practice above and beyond usual time and frequency domain markers.
- The manuscript is not particularly innovative from a methodological standpoint. Since the original study made by M Brennan er al, IEEE Trans Biomed Eng, 48(11):1342-1347, 2001, it is well known that markers derived from Poincaré plot with m=1 are significantly linked to time domain indexes and to frequency domain markers, being SD1 more sensitive to the fast components of heart rate variability and SD2 more sensitive to slow trends and the very low frequency components. While increasing m the dependence of SD1 on spectral markers becomes more complex focusing on progressively slower rhythms but without having a strict correspondence with frequencies expressed in Hz because m is expressed in cardiac beats. Given these considerations the novelty of conclusions should be defended.
- It seems to me that the correspondence between analyses while varying m and spectral indexes holds only in the case of mean heart period equal to 1 s. Please clarify.
- Even from an applicative standpoint the originality of the study should be better emphasized. Several studies analyzed the relationship of Poincaré markers with other heart period variability indexes from an applicative standpoint (see R Maestri et al, J Cardiovasc Electrophysiol, 18, 425-433, 2007; R. Akemi Hoshi et al, Auton Neurosci Basic Clin, 177(2):271-274, 2013). Discussion should be enlarged to account for previous attempts carried out in the past following a less formal approach but very important from an applicative standpoint.
- Conclusions of the study tends to dismiss the independent contribution of Poincaré plot indexes compared to time and frequency domain heart period variability markers that have been stressed by the quantification of the asymmetry of the Poincaré plot (see A. Porta et al, Am J Physiol Reg Integr Comp Physiol, 295, R550-R557, 2008), by the relevance of analysis in higher dimensional embedding spaces (see A. Porta et al, Phys Rev E, 77, 066204, 2008; B. Wang et al, Complexity, 2022, 3880047, 2022) and by the use of alternative methods for the quantification of the Poincaré plot (M Fishman et al, J Appl Physiol, 113(2):297–306, 2012).
Author Response
The study provides an overview of mathematical relations of Poincaré plot markers with time and frequency domain measures of heart period variability.
The study is interesting, even though its originality should be more clearly highlighted and discussion should be enlarged to account for the undoubtful usefulness of Poincaré plot indexes in practice above and beyond usual time and frequency domain markers.
RESPONSE: Although many publications note that SD1 and RMSSD are equivalent, there are still publications in which they are treated as different and empirical correlations between them are calculated. Therefore, we started our manuscript with repeating the findings of Brennan. Also for our novel findings, the mathematics used by Brennan are useful. We hope that more readers can follow our calculations when started with Brennan. We changed a part of the abstract to: “Novel findings are that ln(SD2(m)/SD1(m))=tanh-1(r(m)), which is an approximately normal distributed transformation of r(m) and that SD1(m) and SD2(m) can be calculated by multiplying the power spectrum by a weighing function that depends on m, revealing the relations with spectral measures, but also the relation between SD1(m) and SD2(m).”
The changes we have made are highlighted in turquoise.
The manuscript is not particularly innovative from a methodological standpoint. Since the original study made by M Brennan er al, IEEE Trans Biomed Eng, 48(11):1342-1347, 2001, it is well known that markers derived from Poincaré plot with m=1 are significantly linked to time domain indexes and to frequency domain markers, being SD1 more sensitive to the fast components of heart rate variability and SD2 more sensitive to slow trends and the very low frequency components. While increasing m the dependence of SD1 on spectral markers becomes more complex focusing on progressively slower rhythms but without having a strict correspondence with frequencies expressed in Hz because m is expressed in cardiac beats. Given these considerations the novelty of conclusions should be defended.
RESPONSE: To our knowledge, the transfer function approach is new. The function shows what you describe. For example, it reveals why the contribution of a respiratory peak, that dominates the HF power, to SD1 can disappear or be enhanced for certain m.
It seems to me that the correspondence between analyses while varying m and spectral indexes holds only in the case of mean heart period equal to 1 s. Please clarify.
RESPONSE: Yes, the figures 3 and 4 are only valid for mean IBI=1000 ms. However, the figures can be made for other values of mean IBI, according to equations (39) and (40), but we think the figures become hard to read by adding more graphs for other mean IBI values. That is why we now added mean IBI to the Excel sheets Fig 3 and Fig 4 in the supplement, the graphs can be generated by the reader for a specific mean IBI value and also other values of m. We added to the text and figure legends: “In the supplementary materials, an Excel sheet is available to create these figures for other and m values”.
Even from an applicative standpoint the originality of the study should be better emphasized. Several studies analyzed the relationship of Poincaré markers with other heart period variability indexes from an applicative standpoint (see R Maestri et al, J Cardiovasc Electrophysiol, 18, 425-433, 2007; R. Akemi Hoshi et al, Auton Neurosci Basic Clin, 177(2):271-274, 2013). Discussion should be enlarged to account for previous attempts carried out in the past following a less formal approach but very important from an applicative standpoint.
Conclusions of the study tends to dismiss the independent contribution of Poincaré plot indexes compared to time and frequency domain heart period variability markers that have been stressed by the quantification of the asymmetry of the Poincaré plot (see A. Porta et al, Am J Physiol Reg Integr Comp Physiol, 295, R550-R557, 2008), by the relevance of analysis in higher dimensional embedding spaces (see A. Porta et al, Phys Rev E, 77, 066204, 2008; B. Wang et al, Complexity, 2022, 3880047, 2022) and by the use of alternative methods for the quantification of the Poincaré plot (M Fishman et al, J Appl Physiol, 113(2):297–306, 2012).
RESPONSE: We fully agree that many interesting approaches have been published that, independently, resulted in new measures to characterise the Poincaré plot. Therefore, we now conclude the Discussion as follows: “ We limited our analysis to SD1, SD2, and a number of derived parameters, and did not attempt to review all publications on the subject, as they are so numerous. Clearly, many other approaches to quantify the visual and temporal aspects of Poincaré plots have been developed [19, 28-30]. We think that our findings can be helpful to determine whether (newly) developed measures provide new and independent information.”
Round 2
Reviewer 3 Report
Comments and Suggestions for Authors
The manuscript has been improved. The authors replied satisfactorily to all my issues and took into account the suggestions given. I have no additional comments. I